# Psychometric properties of three online-related addictive behavior instruments among Bangladeshi school-going adolescents

Md. Saiful Islam[1,2]*, Israt Jahan[3], Muhammad Al Amin Dewan[1], Halley M. Pontes[4], Kamrun Nahar Koly[5], Md. Tajuddin Sikder[1], Mahmudur Rahman[1]*

1 Department of Public Health and Informatics, Jahangirnagar University, Savar, Dhaka, Bangladesh, 2 Centre for Advanced Research Excellence in Public Health, Savar, Dhaka, Bangladesh, 3 Institute of Social Welfare and Research, University of Dhaka, Dhaka, Bangladesh, 4 Department of Organizational Psychology, Birkbeck, University of London, London, United Kingdom, 5 Health System and Population Studies Division, icddr,b, Dhaka, Bangladesh

* islam.msaiful@outlook.com (MSI); mahmudur.rahman@juniv.edu (MR)

## Abstract

### Background

Due to the ease of access to the internet in modern society users have become more prone to experiencing addictive behaviors online. The present study aimed to develop and investigate the psychometric properties of the Bangla Internet Gaming Disorder Scale–Short-Form (IGDS9-SF), Gaming Disorder Test (GDT), and Bergen Social Media Addiction Scale (BSMAS) due to a lack of existing sound psychometric tools in Bangladesh.

### Methods

A cross-sectional paper-and-pencil survey was carried out among 428 school-aged adolescents who were active gamers (90.89% males; Mean$_{age}$: 16.13±1.85 years; age range: 10–19 years). Participants were recruited using convenience sampling across four selected schools in Dhaka City, Bangladesh. Data collected included sociodemographic information, frequency of internet use and gaming behaviors, psychological states (i.e., Patient Health Questionnaire [PHQ-9], Generalized Anxiety Disorder [GAD-7]), disordered gaming and social media use (i.e., IGDS9-SF, GDT, and BSMAS). Psychometric testing was conducted to examine the validity and reliability levels of the Bangla IGDS9-SF, GDT, and BSMAS.

### Results

The newly adapted Bangla IGDS9-SF, GDT, and BSMAS exhibited adequate levels of internal consistency. All total scores were significantly correlated with depression, anxiety, frequencies of internet use/online activities and gaming, supporting criterion and convergent validity. CFA indicated excellent construct validity as all instruments had a good fit to the data.

**Data Availability Statement:** All relevant data are within the paper and its Supporting Information files.

**Funding:** The author(s) received no specific funding for this work.

**Competing interests:** The authors have declared that no competing interests exist.

## Conclusions

The findings suggest that the Bangla IGDS9-SF, GDT, and BSMAS are sound psychometric instruments due to their satisfactory psychometric properties including internal consistency, criterion validity, convergent validity, and construct validity.

## Introduction

The latest technological changes and innovations have impacted human civilization worldwide and dramatically changed people's life standards. Technology use and devices help boosting users' communication levels, social group connections, self-reported life satisfaction and well-being levels [1–3]. Despite such positive outcomes, previous studies reported that excessive technology use may lead to detrimental outcomes, with adolescents being considered to be more susceptible to harmful digital technology use [4], particularly in relation to Problematic Internet Use (PIU) [5, 6], with electronic gaming also being a highly popular activity among adolescents due to the latest technological advancements, enticing graphics with realistic images and complex gaming systems [7, 8].

One of the commonly reported negative outcomes of excessive technology use has been related to addictive behavior [9–11]. In terms of gaming behaviors, excessive engagement with the activity can become an addiction upon the experience of functional impairments [12, 13], with a recent large-scale international study of more than 123,000 gamers reporting that disordered gaming may translate to 34.53 to 40.13 hours of weekly time spent gaming [14]. It is widely known that disordered gaming may negatively impact both mental and physical health, cognitive, social, academic, and occupational functioning [7, 15–17]. Disordered gamers display signs of addiction including tolerance, salience, mood-swing, losing control, covering up and risking significant relationships or opportunities [18–21].

Although Gaming Disorder (GD) is now an officially recognized mental health disorder by the World Health Organization (WHO) [22], the American Psychiatric Association (APA) considers it a tentative disorder not formally recognized in the *Diagnostic and Statistical Manual of Mental Disorders* (fifth edition) and adopts 'Internet Gaming Disorder' (IGD) as the official terminology [23]. Disordered gaming has been acknowledged by the APA in the DSM-5 as a condition in need of further study [7, 24]. Nevertheless, the deployment of conflicting terminologies and non-standardized assessment procedures in past research examining disordered gaming [18] has led to multiple debates among researchers as to whether this condition represents a unique clinical entity worth being officially recognized as a behavioral addiction [25–27].

Both GD and IGD are defined as problematic use of electronic games [23], and people showing disordered gaming patterns typically spend a significant amount of time playing video games excessively, which is likely to impair important social relationships, educational achievements, and career opportunities [28]. According to the DSM-5, disordered gaming (i.e., IGD) is defined via nine diagnostic criteria, whereby gamers meeting five or more of these criteria in the last 12 months may be classed as disordered gamers. These nine criteria include: i) preoccupation, ii;) withdrawal symptoms, iii;) tolerance, iv;) difficulty controlling gaming behaviors, v;) loss of interests in other hobbies, vi;) continuation of gaming despite the experience of psychosocial problems, vii;) hiding the amount of gaming, viii;) using games to escape or relieve negative emotions, and ix;) jeopardizing interpersonal, occupational, or educational opportunities due to gaming [29]. At more severe levels, disordered gaming may lead

to academic failure, employment loss or marriage failure because excessive engagement tends to displace expected social, occupational, educational and family related activities and relations [30]. Given the complexity involved in disordered gaming, studies often report an interaction between several internal and external factors, including deficiencies in self, mood, and reward regulation, as well as difficulties with decision-making [31].

According to a systematic review, online gamers exhibit symptoms typically associated with substance abuse, such as mood modification, tolerance, and salience [19], with disordered gaming prevalence rates being about 2.5:1 in favor of males over females [32]. Another systematic review suggested that disordered gaming was associated with a diverse array of personality characteristics, domains, and disorders [33], with a recent study reporting that low conscientiousness and high neuroticism were robustly associated with disordered gaming across both the APA and WHO frameworks [34]. A systematic review assessing disordered gaming cross-sectional and longitudinal epidemiological studies suggested that the prevalence rate of disordered gaming ranged between 0.7% to 27.5%, with males showing a higher prevalence than females while some studies reporting that younger people had a higher prevalence rate than older people [35].

Another relevant phenomenon is related to addictive use of social media platforms. The term 'social media' refers to a variety of internet-based social networks that enable users to communicate verbally and visually with one another [36]. A systematic review examined the evidence for a link between social media use and mental health problems in adolescents, following this research, the most significant risk factors for depression, anxiety, and psychological discomfort were found to be time spent on social media, activities such as repeated checking of messages, and addictive or problematic use of prescription medications [37]. According to a systematic review assessing the impact of social media on mental health [38], there is a significant impact of social media use on an individual's self-perception and mood, as well as their social ties [39].

In terms of assessment, a systematic review of 21 studies investigating 15 language versions of the Internet Gaming Disorder Scale–Short-Form (IGDS9-SF) reported that the scale presented with a unidimensional factor structure, adequate internal consistency, excellent criterion validity, with measurement invariance being supported across gender and age [40]. The item response theory study by Gomez et al. (2019) reported that the IGDS9-SF is a sound psychometric tool that can be used in adolescent and adult gamers as it demonstrated excellent discrimination values [41]. Moreover, [49] suggested an optimal cutoff score of 32 with sensitivity (98.0%) and specificity (91.9%).

Furthermore, several studies involving the analysis of the psychometric properties of disordered gaming tools showed that the Gaming Disorder Test (GDT) is a concise, reliable, and easy to use instrument to assess disordered gaming within a unidimensional factor structure presenting satisfactory reliability, structural validity, and criterion validity [30, 42–46]. Similarly, multiple studies consisting of different language versions of the Bergen Social Media Addiction Scale (BSMAS) reported robust psychometric properties including adequate reliability, structural validity, and criterion validity [47]. Classic test theory and Rasch models demonstrated its appropriateness among adolescent samples [48] and an item response theory and network analysis study also supported these findings among adult samples [47]. Luo et al. (2021) suggested a total score of 24 as the best cut-off score based on the gold standard of a clinical diagnosis with sensitivity (96.4%) and specificity (99.1%) [49].

In Bangladesh, disordered gaming and social media use are increasing at an alarming rate but there is still a lack of research on these two key issues. Students in Bangladesh can easily access the internet and play various online games because of the relatively low cost of internet services and widespread availability of smart devices. Furthermore, about 200 million people

in the world are currently playing PUBG and Free Fire every day [50], with approximately 100 million people downloading those games every day from the Google Play store alone. About 7 million people are playing video games in Bangladesh [50]. Moreover, social media addiction has also become a leading concern among Bangladesh adolescents. Currently, there are about 36 million social media users in Bangladesh, with those aged between 12 to 21 years using social media platforms the most [51].

Several studies mostly conducted among different cohorts including adults from the general population, students, healthcare workers, impoverished urban residents, and COVID-19 survivors reported these common mental health problems (e.g., anxiety, depression, panic, stress, suicidal ideation, and behavioral problems such as problematic use of smartphone, internet, social media) in Bangladesh during the COVID-19 pandemic [52–64]. However, limited research was carried out among adolescents during the COVID-19 in Bangladesh. A recently published study with 322 adolescents reported high levels of depression (67.08%), anxiety (49.38%) and stress (40.68%) in the sample during the COVID-19 pandemic [65]. In Bangladesh, there is currently no robust psychometric tool to assess disordered gaming and social media behaviors. Consequently, the present study aimed to develop and validate three online-related addictive behavior instruments in the Bangladeshi context.

## Methods

### Procedures and participants

A cross-sectional paper-and-pencil survey was conducted among school-aged adolescents from 01 August to 30 September 2021. A convenience sampling technique was employed to recruit all participants from selected schools, leading to an initial sample of 450 participants recruited from five schools in Dhaka City, Bangladesh. The inclusion criteria included: i) being a school-aged adolescent (i.e., 10 to 19 years), ii;) having internet access, iii;) being an active gamer, and iv;) consenting to participating in the survey. Initially, a total of 22 participants had to be removed from the analysis due to having not completed the survey, leading to an effective sample of 428 participants (see 'Statistical analysis and data management strategy section for further information).

All procedures of the present study were performed in accordance with the Institutional Research Ethics guidelines and the Declaration of Helsinki for experiments involving humans. The study protocol was reviewed and approved by the Ethical Review Committee from Uttara Adhunik Medical College, Uttara, Dhaka, Bangladesh. After getting permission from the respective authorities of the selected schools, a data collection date was fixed. Formal written consent was obtained from all participants and their parents prior to data collection, which clearly explained the aims and procedures of the study. Anonymity and confidentiality were granted to all participants and personal identifiable information (e.g., name, telephone number, and address) was not collected in the present study.

### Measures

The survey has collected data pertaining to participants' sociodemographic information, frequency of internet use and gaming behaviors, psychological states (i.e., the adapted versions of the Patient Health Questionnaire [PHQ-9], and Generalized Anxiety Disorder [GAD-7]), disordered gaming and social media use (i.e., IGDS9-SF, GDT, and BSMAS). The IGDS9-SF, GDT, and BSMAS were translated into the Bengali (participants' first language).

For the adaptation of the IGDS9-SF, GDT, and BSMAS, a conceptual translation rather than literal translation was employed to ensure the original meaning of items was preserved while adapting them to the Bangladeshi cultural setting. Three experts fluent in both Bengali

and English reviewed the translated questionnaires. The three instruments were developed adopting the most widely used forward-backward translation guideline [i.e., Beaton et al. 66]. As part of the translation procedure, a modest sample size ($N = 10$) was used in a brief pilot test to assess the scales' readability and legibility.

## Sociodemographic measures

The data regarding sociodemographics included age, gender (male/female), marital status (unmarried/married/in a relationship), academic level (secondary/higher secondary), family type (nuclear/joint), monthly family income, and current living status (with parents/only father or mother/without parents).

## Internet Gaming Disorder Scale–Short-Form (IGDS9-SF)

The IGDS9-SF is a short psychometric tool adapted from the nine criteria defining IGD according to the DSM-5 [23, 67]. This instrument aims to assess the severity of IGD and its detrimental effects by examining both online and/or offline gaming activities occurring over a 12-month period. The nine questions comprising the IGDS9-SF are answered using a five-point Likert scale ranging from 1 ('*never*') to 5 ('*very often*') [67]. The scores are obtained by summing the gamer's answers and total scores can range from 9 to 45, with higher scores being indicative of greater degrees of disordered gaming. Previous research suggested that it may be possible to classify disordered and non-disordered gamers by considering only those gamers that obtain a minimum of 36 out of 45 points in the test [67].

## Gaming Disorder Test (GDT)

The GDT is a brief standardized assessment tool including four items reflecting the key defining diagnostic features of GD in the ICD-11 [42, 68]. The GDT examines gaming activities occurring over a 12-month period since the WHO criteria for GD are based on persistent and recurrent gaming. This most often involves specific online and/or offline games, regardless of the device used to play (e.g., consoles, computers, smartphones). All four items are rated on a five-point Likert scale ranging from 1 ('*never*') to 5 ('*very often*'). Total scores are obtained by summing each item's raw score and range from 4 to 20, with higher scores reflecting higher degrees of disordered gaming [68].

## Bergen Social Media Addiction Scale (BSMAS)

The BSMAS is a short and easy-to-use instrument developed by Andreassen et al. to assess the risk of social media addiction [69]. It was developed based on the six core components of addiction (i.e., salience, mood modification, tolerance, withdrawal conflict, and relapse) proposed by Griffiths [70] and it includes six items rated on a five-point Likert scale ranging from 1 ('*very rarely*') to 5 ('*very often*'). A higher score on the BSMAS suggests a greater risk of social media addiction [69].

## Patient Health Questionnaire (PHQ-9)

Participants' self-reported depression levels were measured using the validated Bangla version of the PHQ-9 [71, 72] which comprises a total of nine items asking participants to rate how they have felt in the previous two weeks. Each question is scored 0 to 3 (0 = '*not at all*', 1 = '*several days*', 2 = '*more than half the days*', and 3 = '*nearly everyday*'). PHQ-9 scores range from 0 to 27, with higher scores indicating higher levels of depression [73].

## Generalized Anxiety Disorder (GAD-7)

Participants' anxiety levels were measured using the Bangla version of the GAD-7, which includes a total of seven items [63, 74]. The GAD-7 items examine: 1) nervousness, 2) inability to stop worrying, 3) excessive worry, 4) restlessness, 5) difficulty in relaxing, 6) easy irritation, and 7) fear of something awful happening. The GAD-7 asks participants to rate how often they have been bothered by each of these seven core anxiety symptoms over the past two weeks. Response categories are '*not at all*', '*several days*', '*more than half the days*', and '*nearly every day*', scored as 0, 1, 2, and 3, respectively. The total score of the GAD-7 ranges from 0 to 21, with the higher scores indicating the higher levels of anxiety [74].

## Statistical analysis and data management strategy

Prior to the main analyses, item-descriptive statistics, univariate outliers, univariate normality, multivariate normality was investigated across all three scales (i.e., IGDS9-SF, GDT, and BSMAS) (see Table 1). In terms of univariate outliers, no participants scored ± 3.29 standard deviations from the total IGDS9-SF, GDT, and BSMAS $z$-scores [75], leading to no additional exclusion of participants. Furthermore, while univariate normality was not violated (i.e., skewness > 3 and kurtosis > 9, see Kline [76]), Mardia's multivariate skewness (3907.804, $p < .001$) and kurtosis (30.077, $p < .001$) coefficients indicated violation of multivariate normality [77]. As such, the Maximum Likelihood Estimation with Robust Standard Errors (MLR) estimator was used to compute the Confirmatory Factor Analysis (CFA) models as it is robust to non-normality and non-independence of observations [78].

**Table 1. Item-descriptive statistics, univariate and multivariate normality testing ($N = 428$).**

| Items | Mean | SD | Min | Max | Skewness | Kurtosis | *W* Statistic | *p*-value |
|---|---|---|---|---|---|---|---|---|
| **Internet Gaming Disorder Scale–Short-Form (IGDS9-SF)** | | | | | | | | |
| IGDS9SF1 | 2.729 | 1.347 | 1 | 5 | 0.327 | -1.062 | 0.888 | < .001 |
| IGDS9SF2 | 2.201 | 1.324 | 1 | 5 | 0.776 | -0.620 | 0.816 | < .001 |
| IGDS9SF3 | 2.173 | 1.291 | 1 | 5 | 0.792 | -0.527 | 0.817 | < .001 |
| IGDS9SF4 | 2.047 | 1.227 | 1 | 5 | 0.890 | -0.293 | 0.796 | < .001 |
| IGDS9SF5 | 1.979 | 1.359 | 1 | 5 | 1.088 | -0.241 | 0.723 | < .001 |
| IGDS9SF6 | 2.262 | 1.381 | 1 | 5 | 0.749 | -0.746 | 0.814 | < .001 |
| IGDS9SF7 | 1.720 | 1.219 | 1 | 5 | 1.521 | 1.023 | 0.638 | < .001 |
| IGDS9SF8 | 2.528 | 1.520 | 1 | 5 | 0.486 | -1.244 | 0.824 | < .001 |
| IGDS9SF9 | 1.820 | 1.305 | 1 | 5 | 1.400 | 0.566 | 0.663 | < .001 |
| **Gaming Disorder Test (GDT)** | | | | | | | | |
| GDT1 | 1.785 | 1.145 | 1 | 5 | 1.311 | 0.661 | 0.711 | < .001 |
| GDT2 | 1.984 | 1.223 | 1 | 5 | 1.096 | 0.167 | 0.775 | < .001 |
| GDT3 | 2.498 | 1.393 | 1 | 5 | 0.438 | -1.118 | 0.858 | < .001 |
| GDT4 | 1.944 | 1.238 | 1 | 5 | 1.066 | -0.115 | 0.754 | < .001 |
| **Bergen Social Media Addiction Scale (BSMAS)** | | | | | | | | |
| BSMAS1 | 2.411 | 1.332 | 1 | 5 | 0.466 | -0.903 | 0.847 | < .001 |
| BSMAS2 | 2.367 | 1.280 | 1 | 5 | 0.551 | -0.712 | 0.857 | < .001 |
| BSMAS3 | 2.621 | 1.421 | 1 | 5 | 0.274 | -1.221 | 0.862 | < .001 |
| BSMAS4 | 2.484 | 1.400 | 1 | 5 | 0.465 | -1.050 | 0.852 | < .001 |
| BSMAS5 | 2.250 | 1.360 | 1 | 5 | 0.723 | -0.738 | 0.817 | < .001 |
| BSMAS6 | 2.493 | 1.441 | 1 | 5 | 0.449 | -1.141 | 0.842 | < .001 |

*Note*: *W* Statistic based on Shapiro-Wilk test.

To assess the quality of structural equation models tested, several fit indices and recommended thresholds were adopted in the present study: χ2/df (1;4), Comparative Fit Index (CFI) and Tucker-Lewis Fit Index (TLI) (0.90–0.95); Root Mean Square Error of Approximation (RMSEA) (0.05;0.08); and the Standardized Root Mean Square Residual (SRMR) (0.05;0.08) [79–82].

All statistical analyses were conducted with R version 4.2.1 (Funny-Looking Kid) (R Core Team, [83]). These included i) descriptive statistics analysis of the sample and main variables, ii) internal consistency analysis (i.e., Cronbach's α, McDonald's ω, and Composite Reliability), iii) CFA to assess construct validity, iii) measurement model comparison via Satorra-Bentler Chi-square difference test for nested models in case of model misfit [84], and iv) correlational analyses with Holm's adjustment method with the main study variables to assess criterion validity and convergent validity. The following R packages were used in the present study to conduct the analyses: psyc v.2.2.5 [85], lavaan v.06-12 (Rosseel, 2012) [86], MVN v.5.9 [87], and semTools v.0.5–6 [88].

## Results

### Characteristics of the participants

Data from a total of 428 participants with a mean age of 16.13 years (SD = 1.85) was analyzed. Of these, most were males ($n$ = 389, 90.89%), unmarried ($n$ = 374, 87.38%), studied in secondary schools ($n$ = 337, 78.74%), and belonged to nuclear families ($n$ = 324, 75.70%). About a third of all participants ($n$ = 149, 34.81%) reported their monthly household income > 45,000 Bangladeshi Taka (BDT) and most reported they lived together with their parents ($n$ = 381, 89.02%). In addition, the average of daily time spent using the internet use and gaming were 6.67 hours (SD = 4.67) and 3.86 hours (SD = 3.50), respectively (see Table 2).

### Internal consistency analysis

In order to assess the internal consistency levels, several reliability coefficients were estimated for the translated IGDS9-SF, GDT, and BSMAS (see Table 3). More specifically, we estimated Cronbach's α, McDonald's ω, and Composite Reliability coefficients. The results of this analysis suggested that the internal consistency of all three scales were satisfactory and beyond the accepted threshold of .70 [89, 90].

### Criterion and convergent validity

To further evaluate the criterion and convergent validity of each translated instrument (i.e., IGDS9-SF, GDT, and BSMAS), Pearson correlation coefficients were calculated for the IGDS9-SF, GDT, BSMAS, PHQ-9, GAD-7, daily time spent on the internet and daily time spent gaming (hours/day) (see Table 4). As expected, IGDS9-SF total scores were significantly and positively correlated with GDT ($r$ = .707), BSMAS ($r$ = .322), PHQ-9 ($r$ = .438), GAD-7 ($r$ = .480), daily time spent on the internet ($r$ = .289), and daily time spent gaming ($r$ = .308).

Furthermore, GDT total scores were significantly and positively correlated with IGDS9-SF ($r$ = .707), BSMAS ($r$ = .249), PHQ-9 ($r$ = .463), GAD-7 ($r$ = .459), daily time spent on the internet ($r$ = .205), and daily time spent gaming ($r$ = .265). Moreover, BSMAS total scores were significantly and positively correlated with IGDS9-SF ($r$ = .322), GDT ($r$ = .249), PHQ-9 ($r$ = .362), GAD-7 ($r$ = .387), daily time spent on the internet ($r$ = .225), and daily time spent gaming ($r$ = .149).

**Table 2. Characteristics of the participants (*N* = 428).**

| *Categorical variables* | *n (%)* |
|---|---|
| **Gender** | |
| Male | 389 (90.89) |
| Female | 39 (9.11) |
| **Marital status** | |
| Unmarried | 374 (87.38) |
| Married | 11 (2.57) |
| In a relationship | 43 (10.05) |
| **Academic level** | |
| Secondary | 337 (78.74) |
| Higher secondary | 91 (21.26) |
| **Family type** | |
| Nuclear | 324 (75.70) |
| Joint | 104 (24.30) |
| **Monthly income** | |
| < 15,000 Bangladeshi Taka (BDT) | 70 (16.36) |
| 15,000–30,000 BDT | 124 (28.97) |
| 30,000–45,000 BDT | 85 (19.86) |
| > 45,000 BDT | 149 (34.81) |
| **Living status** | |
| With parents | 381 (89.02) |
| Only father or mother | 25 (5.84) |
| Without parents | 22 (5.14) |
| *Continuous variables* | *Mean (SD)* |
| Age (year) | 16.13 (1.85) |
| Daily time spent using the internet (hours) | 6.67 (4.67) |
| Daily time spent gaming (hours) | 3.86 (3.50) |
| Depression (score)[a] | 6.29 (6.47) |
| Anxiety (score)[b] | 4.64 (5.39) |

*Note*

[a] Assessed using the Patient Health Questionnaire (PHQ-9)

[b] Assessed using the Generalized Anxiety Disorder (GAD-7)

## Construct validity

Tables 5 and 6 summarize the overall findings of the CFA analyses. In relation to the IGDS9-SF, the CFA results suggested that the one-dimension model had a poor fit to the data ($\chi^2$ = 124.076, df = 27, $p$ < .0001, CFI = 0.873, TLI = 0.831, RMSEA = 0.109, SRMR = 0.063). An inspection of the modification indices suggested that correlating the error variances of

**Table 3. Reliability analysis of the three scales (*N* = 428).**

| Scale | Cronbach's α | McDonald's ω | Composite Reliability |
|---|---|---|---|
| IGDS9-SF | .827 | .827 | .828 |
| GDT | .781 | .785 | .785 |
| BSMAS | .856 | .858 | .859 |

***Abbreviations***: IGDS9-SF: Internet Gaming Disorder Scale–Short-Form; GDT: Gaming Disorder Test; BSMAS: Bergen Social Media Addiction Scale.

**Table 4. Correlational analysis of the main study variables.**

| Variable | 1 | 2 | 3 | 4 | 5 | 6 | 7 |
|---|---|---|---|---|---|---|---|
| IGDS9-SF (1) | 1 | | | | | | |
| GDT (2) | .707 | 1 | | | | | |
| BSMAS (3) | .322 | .249 | 1 | | | | |
| PHQ-9 (4) | .438 | .463 | .362 | 1 | | | |
| GAD-7 (5) | .480 | .459 | .387 | .659 | 1 | | |
| Daily time spent on the internet (6) | .289 | .205 | .225 | .170 | .178 | 1 | |
| Daily time spent gaming (7) | .308 | .265 | .149 | .211 | .194 | .512 | 1 |

*Note*: Pearson correlation coefficients are presented (*p*-values adjusted with Holm's method). All correlation coefficients shown are significant at *p* < .01.

*Abbreviations*: IGDS9-SF: Internet Gaming Disorder Scale–Short-Form; GDT: Gaming Disorder Test; BSMAS: Bergen Social Media Addiction Scale; PHQ-9: Patient Health Questionnaire; GAD-7: Generalized Anxiety Disorder Scale

items IGDS9SF7 with IGDS9SF9 and IGDS9SF1 with IGDS9SF2 would improve model fit. Therefore, a second CFA one-dimensional model was tested, yielding a satisfactory fit to the data ($\chi^2$ = 76.336, df = 25, *p* < .0001, CFI = 0.934, TLI = 0.905, RMSEA = 0.081, SRMR = 0.049), with all standardized factor loadings being statistically significant (ranging from IGDS9SF8 = .471 to IGDS9SF4 = .680). Furthermore, the results of the Satorra-Bentler

**Table 5. Summary of the Confirmatory Factor Analysis results (*N* = 428).**

| Variable | Estimate$_{Model\ 1}$* | SE$_{Model\ 1}$ | Estimate$_{Model\ 2}$* | SE$_{Model\ 2}$ |
|---|---|---|---|---|
| **Internet Gaming Disorder Scale–Short-Form (IGDS9-SF)** | | | | |
| IGDS9SF1 | .527 | - | .516 | - |
| IGDS9SF2 | .501 | 0.099 | .486 | 0.111 |
| IGDS9SF3 | .643 | 0.111 | .664 | 0.120 |
| IGDS9SF4 | .683 | 0.118 | .680 | 0.123 |
| IGDS9SF5 | .642 | 0.149 | .650 | 0.165 |
| IGDS9SF6 | .661 | 0.126 | .676 | 0.141 |
| IGDS9SF7 | .580 | 0.130 | .530 | 0.123 |
| IGDS9SF8 | .474 | 0.118 | .471 | 0.120 |
| IGDS9SF9 | .634 | 0.140 | .592 | 0.131 |
| **Gaming Disorder Test (GDT)** | | | | |
| GDT1 | .600 | - | .659 | - |
| GDT2 | .747 | 0.178 | .622 | 0.116 |
| GDT3 | .726 | 0.205 | .602 | 0.138 |
| GDT4 | .678 | 0.117 | .769 | 0.147 |
| **Bergen Social Media Addiction Scale (BSMAS)** | | | | |
| BSMAS1 | .607 | - | .559 | - |
| BSMAS2 | .769 | 0.074 | .739 | 0.093 |
| BSMAS3 | .698 | 0.100 | .706 | 0.124 |
| BSMAS4 | .748 | 0.104 | .757 | 0.131 |
| BSMAS5 | .708 | 0.096 | .718 | 0.113 |
| BSMAS6 | .716 | 0.099 | .725 | 0.119 |

*Note*

*Statistically significant (i.e., *p* < .001) standardized factor loading estimates.

*Abbreviations*: SE: Robust Standard Error.

**Table 6. Summary of measurement model fit indices across all three scales ($N$ = 428).**

| Model | $\chi^2$ | df | $p$-value | CFI | TLI | RMSEA | RMSEA 90% CI | SRMR | AIC | BIC |
|---|---|---|---|---|---|---|---|---|---|---|
| | | | | *Internet Gaming Disorder Scale–Short-Form (IGDS9-SF)* | | | | | | |
| IGDS9-SF$_{\text{Model 1}}$ | 124.076 | 27 | < .0001 | 0.873 | 0.831 | 0.109 | 0.090–0.129 | 0.063 | 12188.968 | 12262.032 |
| Model 2$_{\text{Model 2}}$ | 76.336 | 25 | < .0001 | 0.934 | 0.905 | 0.081 | 0.061–0.103 | 0.049 | 12123.415 | 12204.598 |
| | | | | *Gaming Disorder Test (GDT)* | | | | | | |
| GDT$_{\text{Model 1}}$ | 19.841 | 2 | < .0001 | 0.955 | 0.864 | 0.158 | 0.099–0.224 | 0.042 | 5171.947 | 5204.420 |
| GDT$_{\text{Model 2}}$ | 1.350 | 1 | < .0001 | 0.999 | 0.993 | 0.027 | 0.000–0.129 | 0.008 | 5151.512 | 5188.044 |
| | | | | *Bergen Social Media Addiction Scale (BSMAS)* | | | | | | |
| BSMAS$_{\text{Model 1}}$ | 26.922 | 9 | < .0001 | 0.972 | 0.953 | 0.085 | 0.049–0.123 | 0.034 | 7957.074 | 8005.784 |
| BSMAS$_{\text{Model 2}}$ | 12.784 | 8 | < .0001 | 0.992 | 0.986 | 0.047 | 0.000–0.092 | 0.021 | 7937.202 | 7989.971 |

*Abbreviations*: df: Degrees of freedom; CFI: Comparative Fit Index; TLI: Tucker-Lewis Index; RMSEA: Root Mean Square Error of Approximation; SRMR: CI: Confidence Interval; Standardized Root Mean Square Residual; AIC: Akaike Information Criterion; BIC: Bayesian Information Criterion.

Chi-Square ($\chi^2$) difference test for nested models indicated that in the case of the IGDS9-SF, Model 2 presented with significantly better fit to the data than Model 1 after correlating the suggested error variances ($\triangle\chi^2$ = 39.306, $\triangle$df = 2. $p$ < .0001).

As for the GDT, the CFA results indicated that the one-dimension model also presented with poor fit to the data ($\chi^2$ = 19.841, df = 2, $p$ < .0001, CFI = 0.955, TLI = 0.864, RMSEA = 0.158, SRMR = 0.042). After inspecting the modification indices, it was suggested that correlating the error variances of items GDT2 with GDT3 would lead to fit improvement. Following this, a second CFA one-dimensional model was estimated, yielding excellent fit to the data ($\chi^2$ = 1.350, df = 1, $p$ < .0001, CFI = 0.999, TLI = 0.993, RMSEA = 0.027, SRMR = 0.008), with all standardized factor loadings being statistically significant (ranging from GDT3 = .602 to GDT4 = .769). Additionally, the results of the Satorra-Bentler Chi-Square ($\chi^2$) difference test for nested models indicated that in the case of the GDT, Model 2 presented with significantly better fit to the data than Model 1 after correlating the suggested error variances ($\triangle\chi^2$ = 15.107, $\triangle$df = 1. $p$ < .0001).

In regards to the BSMAS, the results of the CFA suggested that the one-dimension model presented with a relatively modest fit to the data ($\chi^2$ = 26.922, df = 9, $p$ < .0001, CFI = 0.972, TLI = 0.953, RMSEA = 0.085, SRMR = 0.034). However, upon inspection of the modification indices, it was found that correlating the error variances of items BSMAS1 with BSMAS2 would result in improved fit. Thus, a second CFA one-dimensional model was fitted and the results indicated excellent fit to the data ($\chi^2$ = 12.784, df = 8, $p$ < .0001, CFI = 0.992, TLI = 0.986, RMSEA = 0.047, SRMR = 0.021), with all standardized factor loadings being statistically significant (ranging from BSMAS1 = .559 to BSMAS4 = .757). Similarly, to the previous findings, the results of the Satorra-Bentler Chi-Square ($\chi^2$) difference test for nested models also indicated that Model 2 presented with significantly better fit to the data than Model 1 after correlating the suggested error variances ($\triangle\chi^2$ = 14.822, $\triangle$df = 1. $p$ < .0001).

## Discussion

The present study aimed to develop and investigate the psychometric properties, including internal consistency, criterion validity, convergent validity, and construct validity of three widely used online-related addictive behavior instruments (i.e., IGDS9-SF, GDT, and BSMAS) among Bangladeshi school-aged adolescents. The findings of the present study are consistent with previous studies which also demonstrated that the IGDS9-SF [67], GDT [42], and BSMAS [69] are sound psychometric instruments.

The IGDS9-SF demonstrated adequate levels of internal consistency in the current study, which is comparable to the findings of a systematic review involving 21 studies comprising 15 language versions of the IGDS9-SF, further supporting the methodological quality and a positive rating for the quality of statistical findings concerning its internal consistency [40]. In the present study, all fit indices obtained for Model 2 tested in the CFA were satisfactory and within acceptable conventional thresholds, which is also consistent with previous findings [40] and supports the scale's construct validity. The IGDS9-SF was significantly correlated with GDT, frequency of internet use and frequency of gaming, further supporting both the criterion and convergent validity of the IGDS9-SF, which further corroborates previous studies [9, 67, 91–97] as the frequency of gaming is related to the severity of disordered gaming. Additionally, IGDS9-SF scores were significantly correlated with depression and anxiety levels, a finding that converges with that of previous research [93, 95, 97]. In addition, IGDS9-SF scores were significantly correlated with the BSMAS scores, which was also observed in the literature [91, 92, 97–99].

As for the adapted GDT, the results showed that the scale exhibited acceptable internal consistency levels in the current study [89, 90]. This finding is aligned with previous studies [30, 42–46]. In the present study, all fit indices obtained for Model 2 tested in the CFA were adequate and within their conventional thresholds, further supporting the scale's unidimensional structure and converging with results reported in previous research [30, 42–46] while also supporting the scale's construct validity. Furthermore, GDT scores were significantly and positively correlated with IGDS9-SF scores, frequency of internet use and frequency of gaming, supporting both the criterion and convergent validity of the tool and corroborating previous studies [42–45]. The findings obtained also suggested that the frequency of gaming was positively related to the severity of IGD. Moreover, GDT scores were significantly and positively correlated with depression and anxiety levels. Previous studies also indicated that disordered gaming was significantly associated with poor mental states (e.g., depression, anxiety, etc.) [43, 93, 95, 97]. In addition, GDT scores were significantly and positively correlated with BSMAS scores, supporting previous research reporting that disordered gaming levels were also positively correlated with social media use [91, 92, 97].

The translated and adapted Bangla version of the BSMAS also had excellent internal consistency in the current study, which is in line with previous research [69, 91, 92, 97, 100, 101]. In the present study, all fit indices obtained for Model 2 tested in the CFA were adequate and within their conventional thresholds, also supporting the scale's unidimensional structure, which is consistent with the previous studies [48, 69, 91, 92, 97, 100] and supports the scale's construct validity. Similarly, the BSMAS was significantly and positively correlated with online behaviors including the frequency of internet use and frequency of gaming, supporting both criterion and convergent validity as has been reported in previous research [91, 97]. Furthermore, BSMAS scores were significantly and positively correlated with depression and anxiety levels, further supporting the findings of previous studies [48, 69, 97]. In addition, BSMAS scores were significantly and positively correlated with the IGDS9-SF and GDT scores. These findings are aligned with the results of prior studies suggesting that social media use was also correlated with IGD [48, 91, 92, 97].

## Limitations

The present study included several potential limitations that should be considered when interpreting the findings reported. First, this study was conducted with a modest sample of adolescents from specific secondary and higher secondary schools in Dhaka. As a result, the findings may not be generalizable to other populations, geographic areas, or indeed other nations. To

overcome this issue, studies using random sampling with larger samples would be beneficial in producing more generalizable findings. Second, since this was a cross-sectional study, causality in the findings reported cannot assumed. Moreover, further investigations are warranted with the inclusion of incremental measures along with their application to clinical settings in clinically-diagnosed participants.

## Conclusions

The findings indicated that the newly adapted Bangla versions of the IGDS9-SF, GDT, and BSMAS were robust instruments since they all met the criteria of reliability, criterion validity, convergent validity, and construct validity. Consequently, the Bangla versions of the IGDS9-SF, GDT, and BSMAS provide a convenient and quick method for screening online-related addictive behavior in Bangladesh in a sound way. These instruments will help the researchers and/or clinicians to contribute in future epidemiological studies or clinical settings.

## Supporting information

**S1 File. Data set.**
(XLSX)

**S2 File. The correlations and associations of the three scales (e.g., IGDS9-SF, GDT, and BSMAS) with socio-demographic variables.**
(DOCX)

## Acknowledgments

The authors would like to express the most profound gratitude to all of the respondents who participated in this study.

## Author Contributions

**Conceptualization:** Md. Saiful Islam, Israt Jahan, Muhammad Al Amin Dewan, Halley M. Pontes, Kamrun Nahar Koly, Md. Tajuddin Sikder, Mahmudur Rahman.

**Data curation:** Israt Jahan, Muhammad Al Amin Dewan.

**Formal analysis:** Md. Saiful Islam, Halley M. Pontes.

**Investigation:** Md. Saiful Islam, Israt Jahan, Muhammad Al Amin Dewan.

**Methodology:** Md. Saiful Islam, Israt Jahan, Muhammad Al Amin Dewan.

**Supervision:** Mahmudur Rahman.

**Validation:** Md. Saiful Islam, Israt Jahan, Muhammad Al Amin Dewan, Halley M. Pontes, Kamrun Nahar Koly, Md. Tajuddin Sikder, Mahmudur Rahman.

**Writing – original draft:** Md. Saiful Islam, Israt Jahan, Muhammad Al Amin Dewan, Halley M. Pontes.

**Writing – review & editing:** Md. Saiful Islam, Halley M. Pontes, Kamrun Nahar Koly, Md. Tajuddin Sikder, Mahmudur Rahman.

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
