## [Decision Letter · Decision Letter 0]

11 Aug 2022

PONE-D-22-06872Psychometric properties of three online-related addictive behavior instruments among Bangladeshi school-going adolescentsPLOS ONE

Dear Dr. Islam,

Thank you for submitting your manuscript to PLOS ONE. After careful consideration, we feel that it has merit but does not fully meet PLOS ONE’s publication criteria as it currently stands. Therefore, we invite you to submit a revised version of the manuscript that addresses the points raised during the review process.

The manuscript has been evaluated by two reviewers, and their comments are available below.

The reviewers have raised a number of major concerns. They request improvements to the reporting of methodological aspects of the study such as clarification regarding the data set and analysis, and the questionnaires used.

Could you please carefully revise the manuscript to address all comments raised?

We look forward to receiving your revised manuscript.

Kind regards,

Lorena Verduci

Staff Editor

PLOS ONE

Journal Requirements:

Reviewers' comments:

Reviewer's Responses to Questions

**Comments to the Author**

1. Is the manuscript technically sound, and do the data support the conclusions?

Reviewer #1: Yes

Reviewer #2: Partly

2. Has the statistical analysis been performed appropriately and rigorously? 

Reviewer #1: Yes

Reviewer #2: No

3. Have the authors made all data underlying the findings in their manuscript fully available?

Reviewer #1: Yes

Reviewer #2: Yes

4. Is the manuscript presented in an intelligible fashion and written in standard English?

Reviewer #1: Yes

Reviewer #2: Yes

5. Review Comments to the Author

Reviewer #1: Thank-you for the opportunity to review the manuscript, “Psychometric properties of three online-related addictive behavior instruments among Bangladeshi school-going adolescents”. The purpose of the study was to develop and investigate the psychometric properties of the Bangla Internet Gaming Disorder Scale– Short-Form (IGDS9-SF), Gaming Disorder Test (GDT), and Bergen Social Media Addiction Scale (BSMAS). The study adopted a cross-sectional survey with paper-and-pencil. While the manuscript was generally well-written, I have some concerns about key study information that appears to be missing from the manuscript; and, some clarification from the authors regarding the data set and analyses are also recommended. My comments are outlined below and I sincerely hope that the authors find them helpful in any future revisions of their work.

1. In the introduction section, I suggest that the authors describe some to the three main scales of the article. A brief review of the research on these three scales in relation to the measurement of Internet Gaming Disorderm, Gaming Disorder and social media addiction and clarification of the importance of these three scales.

2. In the participants section, due to the global impact of COVID-19 and the possible impact of the pandemic on problematic Internet behavior among adolescents, I suggest that the authors include a description of the pandemic in Bangladesh (with particular emphasis on the impact on adolescents)

3. Regarding to the demographic measure, it is curious why included maritial status, family type, monthly family income, and current living status? From the following analysis, these demographic variables did not have any relationship with the statistical analysis. Perhaps the authors could consider further comparisons for these personal background variables on the three scales (presented in an appendix).

4. In the results section, I suggest that the authors use the Shapiro-Wilk test to test whether the data conform to a normal distribution, rather than just presenting the values of Skewness and Kurtosis. In the case of addiction measures, the data can easily violate the normal distribution, so perhaps using other estimation (e.g., DWLS) for CFA is a better way to address this issue.

5. I also suggested to explain why adding the estimation of the correlation of measurement error of the items in Figure 1.

6. Regarding to the convergent validity, it's a bit confusing, why the correlation coefficient between the scales can demonstrate the convergent validity. Moreover, there is also a demonstration of the convergent validity in the section of scale-level psychometric properties. It is recommended that the author add explanations or adjust the positions of the two.

Reviewer #2: The presentation of the validation of the adapted versions of three scales belonging to the topic of internet addiction is to be appreciated. Beyond these, there are a number of methodological issues, more precisely certain aspects that do not consistently support what the authors claim. Find the related details below:

(1) The theoretical framing of the research is missing. The relevant theories for the behavioral addictions analyzed in this research should be mentioned.

(2) Previous studies in the psychometric literature regarding versions adapted in other languages/cultures are also missing. A few are only mentioned in the Discussions section. To complete these aspects and to emphasize the importance/necessity of the extension of this empirical body, an extension to which you are actually trying to contribute.

(3) It should also be mentioned that the validation studies include CTT, as well as advanced approaches (i.e., IRT).

For IGD9SF mention:

https://doi.org/10.1007/s11469-018-9890-z

For BSMAS mention:

https://doi.org/10.1556/2006.6.2017.071

https://doi.org/10.1007/s11469-021-00732-7

Also mention the studies that proposed various cutoffs:

For IGD9SF

https://doi.org/10.3389/fpsyt.2020.00470

For BSMAS

https://doi.org/10.1556/2006.2021.00025

Even if you mentioned some of them (not all) in the Discussions, they must be specified (synthetically, obviously) in the theoretical background as well

(4) Page 8, line 3: It should be mentioned that the adapted versions of the Patient Health Questionnaire [PHQ-9] and Generalized Anxiety Disorder [GAD-7] questionnaires were used).

(5) You did not mention anything about outliers, multivariate data distribution, estimation method used, in CFA

(6) You analyzed the reliability using the Cronbach alpha coefficient, although the tau-equivalence assumption is clear that it is not fulfilled because all item discriminations are not equal. To be completed here with model-based reliability, which does not require tau-equivalence

(7) You did not explain the connection between floor effect and ceiling effect with psychometric robustness. More precisely, you emphasized that these effects are not present (although in the case of one scale the cutoff recommended in the literature was slightly exceeded), but you did not explain which is relevant to the validity of the studied scales

(8) According to the data presented in Table 6, it is a problem related to the IGD9SF scale. From what you report, it appears that only 34% of the variance is explained by the modeled latent variable, that is, internet gaming disorder.

You did not correctly reproduce the amendment proposed by Fornell and Larker (1981). They specified that AVE has a cutoff >.50, but a threshold of .40 is accepted only if CR>.60. You mentioned something else on page 15, namely that it is about values lower than .50, although the authors explicitly mention that the threshold of .40 is also accepted. It can be seen that this criterion is not met in your case, since AVE is .34.

(9) You did not discuss anything about the fact that a third of the items of this scale have small slopes, slightly above the allowed threshold. Obviously, that is why the AVE is so small.

(10) You correlated the residuals in three situations and did not explain anything. Anyway, this option is considered in the psychometric literature to be avoided because it represents an artificial forcing to obtain a good model fit.

To inspect the covariance matrix

(11) It should be mentioned that the RMSEA in the case of BSMAS indicates a mediocre fit between your model and the data

(12) Page 17 _ You stated that: "In the present study, all fit indices obtained in the CFA were well-accepted in their conventional thresholds". As I mentioned above, it is to be discussed here. The term “all” is too inclusive compared to the results obtained. So reformulate, nuance and explain.

Good luck!

6. PLOS authors have the option to publish the peer review history of their article (what does this mean?). If published, this will include your full peer review and any attached files.

Reviewer #1: No

Reviewer #2: **Yes: **Elena Stanculescu

---

## [Author Response · Author response to Decision Letter 0]

30 Sep 2022

Dear Dr. Lorena Verduci,

We would like to thank you for your consideration of our manuscript ‘Psychometric properties of three online-related addictive behavior instruments among Bangladeshi school-going adolescents’ and for the feedback provided by the reviewers. We have now addressed each comment provided by the reviewers and we hope that the revised paper is now acceptable for publication in the PLOS ONE.

Responses to Reviewer 1

Reviewer’s comment: Thank-you for the opportunity to review the manuscript, “Psychometric properties of three online-related addictive behavior instruments among Bangladeshi school-going adolescents”. The purpose of the study was to develop and investigate the psychometric properties of the Bangla Internet Gaming Disorder Scale– Short-Form (IGDS9-SF), Gaming Disorder Test (GDT), and Bergen Social Media Addiction Scale (BSMAS). The study adopted a cross-sectional survey with paper-and-pencil. While the manuscript was generally well-written, I have some concerns about key study information that appears to be missing from the manuscript; and, some clarification from the authors regarding the data set and analyses are also recommended. My comments are outlined below and I sincerely hope that the authors find them helpful in any future revisions of their work.

Authors’ response: Thanks for your review and feedback. We have now revised the manuscript as you suggested.

Reviewer’s comment: In the introduction section, I suggest that the authors describe some to the three main scales of the article. A brief review of the research on these three scales in relation to the measurement of Internet Gaming Disorderm, Gaming Disorder and social media addiction and clarification of the importance of these three scales.

Authors’ response: We have now revised the Introduction and added a brief review of three scales as suggested. We hope the added literature and discussion will improve the manuscript. 

Reviewer’s comment: In the participants section, due to the global impact of COVID-19 and the possible impact of the pandemic on problematic Internet behavior among adolescents, I suggest that the authors include a description of the pandemic in Bangladesh (with particular emphasis on the impact on adolescents)

Authors’ response: Thanks for the important suggestion. We have expanded the study rationale and added relevant COVID-19 literature in the Introduction.

Reviewer’s comment: Regarding to the demographic measure, it is curious why included maritial status, family type, monthly family income, and current living status? From the following analysis, these demographic variables did not have any relationship with the statistical analysis. Perhaps the authors could consider further comparisons for these personal background variables on the three scales (presented in an appendix).

Authors’ response: Thanks for your suggestion. We have now included the correlations and associations of the three scales with socio-demographic variables. As we are also making the data set of the study publicly available, any interested researcher can run additional analyses if they wish to do so.

Reviewer’s comment: In the results section, I suggest that the authors use the Shapiro-Wilk test to test whether the data conform to a normal distribution, rather than just presenting the values of Skewness and Kurtosis. In the case of addiction measures, the data can easily violate the normal distribution, so perhaps using other estimation (e.g., DWLS) for CFA is a better way to address this issue.

Authors’ response: We thank the reviewer for the important comment, which led us to re-run all the analyses and re-write the findings, which did not change substantially compared to the previous version of the manuscript. The reviewer will see that we have now added a clearer rationale to our data management and statistical analysis approach. This is described in the Methods section, namely the ‘Procedures and participants’ and ‘Statistical analysis and data management strategy’ subsections. The new information reports our investigation of outliers and normality. As we did not find a consistent normal distribution in the data, we used the Maximum Likelihood Estimation with Robust Standard Errors (MLR) estimator to compute the Confirmatory Factor Analysis (CFA) models as it is robust to non-normality and non-independence of observations and is recommended by Muthén and Muthén (2018). The new results refer to robust estimates where applicable.

Reviewer’s comment: I also suggested to explain why adding the estimation of the correlation of measurement error of the items in Figure 1.

Authors’ response: Thanks for the important comment. Due to the use of the new estimator, the CFA results were slightly different to the previous ones. As such, we opted to investigate modification indices in order to address potential model misfit, which is a common practice in the field. The reviewer will see now that in cases where model misfit was detected, a second model, clearly explaining which error variances were correlated, was then computed and tested against the first model using Satorra-Bentler Chi-square different test for nested models so that the choice of the best-fitting model is empirically supported. We hope the new information is clearer and better presented.

Reviewer’s comment: Regarding to the convergent validity, it's a bit confusing, why the correlation coefficient between the scales can demonstrate the convergent validity. Moreover, there is also a demonstration of the convergent validity in the section of scale-level psychometric properties. It is recommended that the author add explanations or adjust the positions of the two.

Authors’ response: Thanks for the comment. The new analyses led to additional information and changes to the Results section as explained in the previous comment. We focused the discussion regarding the three scales’ validity in terms of construct validity, criterion validity, and convergent validity to be clearer. Each type of validity is supported by different analyses, which we also explain now in the revised Statistical analysis and data management strategy subsection within the Methods. This said, we tested construct validity via CFA, then criterion validity and convergent validity were tested by examining how each scale’s total scores are associated with related criterion variables and measures that the literature suggests some degree of convergence. We hope the new information has improved the manuscript.

 

Responses to Reviewer 2

Reviewer’s comment: The presentation of the validation of the adapted versions of three scales belonging to the topic of internet addiction is to be appreciated. Beyond these, there are a number of methodological issues, more precisely certain aspects that do not consistently support what the authors claim. Find the related details below:

Authors’ response: Thanks for your review and feedback. We have now addressed your comments and revised the manuscript accordingly. 

Reviewer’s comment: The theoretical framing of the research is missing. The relevant theories for the behavioral addictions analyzed in this research should be mentioned.

Authors’ response: We thank you for the comment provided. We have made significant revisions to the manuscript, including adding more literature to the Introduction.

Reviewer’s comment: Previous studies in the psychometric literature regarding versions adapted in other languages/cultures are also missing. A few are only mentioned in the Discussions section. To complete these aspects and to emphasize the importance/necessity of the extension of this empirical body, an extension to which you are actually trying to contribute. It should also be mentioned that the validation studies include CTT, as well as advanced approaches (i.e., IRT).

For IGD9SF mention:

https://doi.org/10.1007/s11469-018-9890-z

For BSMAS mention:

https://doi.org/10.1556/2006.6.2017.071

https://doi.org/10.1007/s11469-021-00732-7

Also mention the studies that proposed various cutoffs:

For IGD9SF

https://doi.org/10.3389/fpsyt.2020.00470

For BSMAS

https://doi.org/10.1556/2006.2021.00025

Even if you mentioned some of them (not all) in the Discussions, they must be specified (synthetically, obviously) in the theoretical background as well

Authors’ response: All references have now been cited and discussed in the manuscript. 

Reviewer’s comment: Page 8, line 3: It should be mentioned that the adapted versions of the Patient Health Questionnaire [PHQ-9] and Generalized Anxiety Disorder [GAD-7] questionnaires were used).

Authors’ response: This has now been modified as suggested. 

Reviewer’s comment: You did not mention anything about outliers, multivariate data distribution, estimation method used, in CFA

Authors’ response: Given the concerns raised by Reviewer 1, we have now re-run all the analyses. In the revised version of the manuscript, the issue of outliers and normality is now tested and discussed in the revised Methods section.

Reviewer’s comment: You analyzed the reliability using the Cronbach alpha coefficient, although the tau-equivalence assumption is clear that it is not fulfilled because all item discriminations are not equal. To be completed here with model-based reliability, which does not require tau-equivalence

Authors’ response: Thanks for your comments. We have now focused our internal consistency analyses around three reliability coefficients: Cronbach’s alpha, McDonald’s omega, and Composite Reliability. The additional reliability coefficients help overcoming the important shortcomings of Cronbach’s alpha as importantly raised by the reviewer.

Reviewer’s comment: You did not explain the connection between floor effect and ceiling effect with psychometric robustness. More precisely, you emphasized that these effects are not present (although in the case of one scale the cutoff recommended in the literature was slightly exceeded), but you did not explain which is relevant to the validity of the studied scales

Authors’ response: Due to the fact that we had to re-run all the analyses as per the previous comments by Reviewer 1, we have opted to focus the validation process in terms of more standard metrics such as reliability and validity at multiple levels. We are sharing our data set with all interested readers should they wish to conduct further analyses.

Reviewer’s comment: According to the data presented in Table 6, it is a problem related to the IGD9SF scale. From what you report, it appears that only 34% of the variance is explained by the modeled latent variable, that is, internet gaming disorder.

You did not correctly reproduce the amendment proposed by Fornell and Larker (1981). They specified that AVE has a cutoff >.50, but a threshold of .40 is accepted only if CR>.60. You mentioned something else on page 15, namely that it is about values lower than .50, although the authors explicitly mention that the threshold of .40 is also accepted. It can be seen that this criterion is not met in your case, since AVE is .34.

Authors’ response: Similarly to the last comment, we believe this issue is not relevant anymore as we now focus more on the key internal consistency and validity aspects of the new scales in the revised manuscript.

Reviewer’s comment: You did not discuss anything about the fact that a third of the items of this scale have small slopes, slightly above the allowed threshold. Obviously, that is why the AVE is so small.

Authors’ response: Kindly see our response to the comment above.

Reviewer’s comment: You correlated the residuals in three situations and did not explain anything. Anyway, this option is considered in the psychometric literature to be avoided because it represents an artificial forcing to obtain a good model fit.

Authors’ response: We have now reported this aspect of the analysis more clearly and transparently in the revised Methods.

Reviewer’s comment: It should be mentioned that the RMSEA in the case of BSMAS indicates a mediocre fit between your model and the data

Authors’ response: Due to the new estimator used in the CFAs, we have now obtained slightly different results and we have also conducted additional analyses to investigate the fit of the models tested. The new information is now reported in the revised Results section.

Reviewer’s comment: Page 17 _ You stated that: “In the present study, all fit indices obtained in the CFA were well-accepted in their conventional thresholds”. As I mentioned above, it is to be discussed here. The term “all” is too inclusive compared to the results obtained. So reformulate, nuance and explain.

Good luck!

Authors’ response: We have now added more specificity to our model fit analysis and interpretation by specifying the thresholds we used. We have also revised that particular passage of the text in the revised manuscript.

---

## [Decision Letter · Decision Letter 1]

1 Dec 2022

Psychometric properties of three online-related addictive behavior instruments among Bangladeshi school-going adolescents

PONE-D-22-06872R1

Dear Dr. Islam,

We’re pleased to inform you that your manuscript has been judged scientifically suitable for publication and will be formally accepted for publication once it meets all outstanding technical requirements.

Kind regards,

Keiko Nakamura

Academic Editor

PLOS ONE

Additional Editor Comments (optional):

Reviewers' comments:

Reviewer's Responses to Questions

**Comments to the Author**

1. If the authors have adequately addressed your comments raised in a previous round of review and you feel that this manuscript is now acceptable for publication, you may indicate that here to bypass the “Comments to the Author” section, enter your conflict of interest statement in the “Confidential to Editor” section, and submit your "Accept" recommendation.

Reviewer #1: All comments have been addressed

Reviewer #2: (No Response)

2. Is the manuscript technically sound, and do the data support the conclusions?

Reviewer #1: Yes

Reviewer #2: (No Response)

3. Has the statistical analysis been performed appropriately and rigorously? 

Reviewer #1: Yes

Reviewer #2: (No Response)

4. Have the authors made all data underlying the findings in their manuscript fully available?

Reviewer #1: Yes

Reviewer #2: (No Response)

5. Is the manuscript presented in an intelligible fashion and written in standard English?

Reviewer #1: Yes

Reviewer #2: (No Response)

6. Review Comments to the Author

Reviewer #1: Thanks to the author for revising the manuscript. The questions I raised have been properly answered and corrected by the authors. The quality of the article has also improved. I suggest that this article can be published, and good luck to the author

Reviewer #2: (No Response)

7. PLOS authors have the option to publish the peer review history of their article (what does this mean?). If published, this will include your full peer review and any attached files.

Reviewer #1: **Yes: **I-Hua Chen

Reviewer #2: **Yes: **Elena Stănculescu

---

## [Editor Report · Acceptance letter]

4 Dec 2022

PONE-D-22-06872R1 

Psychometric properties of three online-related addictive behavior instruments among Bangladeshi school-going adolescents 

Dear Dr. Islam:

I'm pleased to inform you that your manuscript has been deemed suitable for publication in PLOS ONE. Congratulations! Your manuscript is now with our production department. 

Kind regards, 

on behalf of

Professor Keiko Nakamura 

Academic Editor

PLOS ONE